

# A multi-ethnic proteomic profiling analysis in Alzheimer's disease identifies the disparities in dysregulation of proteins and pathogenesis

Mei Sze Tan[1], Phaik-Leng Cheah[2], Ai-Vyrn Chin[3], Lai-Meng Looi[2] and Siow-Wee Chang[1,4]

[1] Bioinformatics Programme, Institute of Biological Sciences, Faculty of Science, Universiti Malaya, Kuala Lumpur, Malaysia
[2] Department of Pathology, Faculty of Medicine, Universiti Malaya, Kuala Lumpur, Malaysia
[3] Department of Medicine, Faculty of Medicine, Universiti Malaya, Kuala Lumpur, Malaysia
[4] Centre of Research in System Biology, Structural, Bioinformatics and Human Digital Imaging (CRYSTAL), Universiti Malaya, Kuala Lumpur, Malaysia

Corresponding author
Siow-Wee Chang,
siowwee@um.edu.my

## ABSTRACT

**Background:** Alzheimer's disease (AD) is the most common type of dementia that affects the elderly population. Lately, blood-based proteomics have been intensively sought in the discovery of AD biomarkers studies due to the capability to link external environmental factors with the development of AD. Demographic differences have been shown to affect the expression of the proteins in different populations which play a vital role in the degeneration of cognitive function.
**Method:** In this study, a proteomic study focused on Malaysian Chinese and Malay prospects was conducted. Differentially expressed proteins (DEPs) in AD patients and normal controls for Chinese and Malays were identified. Functional enrichment analysis was conducted to further interpret the biological functions and pathways of the DEPs. In addition, a survey investigating behavioural practices among Chinese and Malay participants was conducted to support the results from the proteomic analysis.
**Result:** The variation of dysregulated proteins identified in Chinese and Malay samples suggested the disparities of pathways involved in this pathological condition for each respective ethnicity. Functional enrichment analysis supported this assumption in understanding the protein-protein interactions of the identified protein signatures and indicate that differentially expressed proteins identified from the Chinese group were significantly enriched with the functional terms related to Aβ/tau protein-related processes, oxidative stress and inflammation whereas neuroinflammation was associated with the Malay group. Besides that, a significant difference in sweet drinks/food intake habits between these two groups implies a relationship between sugar levels and the dysregulation of protein *APOA4* in the Malay group. Additional meta-analysis further supported the dysregulation of proteins *TF, AHSG, A1BG, APOA4* and *C4A* among AD groups.
**Conclusion:** These findings serve as a preliminary understanding in the molecular and demographic studies of AD in a multi-ethnic population.

## INTRODUCTION

Alzheimer's disease (AD) is the most common type of dementia, an age-related neurodegenerative disease characterized by unusual changes in the brain which subsequently cause devolution in behaviour, thinking, memory and ability to carry out daily tasks (*Tan et al., 2021*). Several risk factors are reported to be associated with AD, such as age, hereditary and family history (*Alzheimer's Association, 2020*; *Mucke, 2009*). Several proteins are known to be associated with the initiation and development of AD, including β-amyloid (*Aβ*), tau-protein, amyloid precursor protein (*APP*) and ε4 allele of apolipoprotein E (*ApoE4*). Although numerous studies have been performed to explain AD's aetiology, most of the findings were related to *Aβ* plaque and tau tangle deposition in the brain. The altered and causative mechanism, together with the chemical and molecular components that lead to AD pathology remain unclear.

Currently, the clinical diagnosis of AD mainly depends on a series of examinations based on neurophysiological assessments of the patient's cognitive function (*Daffner, 2000*; *Yao et al., 2019*). A variety of techniques have been widely used recently to support the diagnosis of AD such as neuroimaging techniques *e.g.*, positron emission tomography (PET) and neurochemical assay testing in the cerebrospinal fluid (CSF) and blood (*Shen et al., 2017*). As blood is more accessible than CSF, the investigation of blood-based biomarkers is gaining traction in AD studies lately, as an alternative to CSF and PET examinations that are rather invasive or high in price (*Zetterberg & Burnham, 2019*). Several evaluations have been done using blood-based biomarkers in AD, but the results have been inconsistent (*Janelidze et al., 2016*; *Rehiman et al., 2020a*, *2020b*; *Yao et al., 2019*). Such inconsistencies observed in *Aβ, APP* and *ApoE4* could be due to their low concentrations in blood compared with CSF as well as the variations in the methods used (*Zetterberg & Burnham, 2019*). Nevertheless, we feel blood for assay of biomarkers, due to its easy accessibility, will remain an important candidate for studying AD, and hence our interest in developing along this line, with augmentation of standardised technical methods by advanced bioinformatics.

Large-scale genomic and transcriptomic studies have been carried out to uncover the disease pathological networks and their related novel therapeutic markers (*Jansen et al., 2019*; *Patel, Dobson & Newhouse, 2019*; *Su et al., 2019*; *Wightman et al., 2020*); however, these have not been able to significantly indicate the functional gene products *i.e.*, proteins (*Bai et al., 2020*; *Liu, Beyer & Aebersold, 2016*). On the other hand, the set of involved proteins, known as proteomes, is more dynamic than what can be captured by genomics and transcriptomics, as protein expressions are often influenced by external environmental factors (*International Service for the Acquisition of Agri-biotech Applications, 2006*). The identification of aberrant protein expressions that affect the pathogenesis of AD should still be the main goal in studying the perturbation of processes related to AD (*Madrid et al., 2021*; *Shen et al., 2017*; *Zahid et al., 2014*). With the advancement in technology, for

example, by using mass spectrometry (MS), there can be in-depth profiling and quantification of proteins in biological samples. The relatively high resolution of MS technology has replaced the application of the low resolution and time-consuming two-dimensional gel electrophoresis (2-DE) in measuring protein information, especially in complex diseases (Tan et al., 2021). Computational methods to analyse a large number of extracted proteins have been crucial to provide further insight.

There are many factors involved in affecting the regulation of proteins, such as genetic regulation, genetic coding and external environmental factors (Wu et al., 2021). Among these factors, the correlation between the environmental factors (i.e., living environment, lifestyle practices and dietary plans) with the dysregulation of proteins in AD is the main interest of this study.

Malaysia has a multi-ethnic population that comprises Malays (69.6%), Chinese (22.6%), Indians (6.8%) and other races (1.0%) (Department of Statistics Malaysia, 2021). In the same year, people ≥65 years of age formed about 7.4% of the total 32.7 million Malaysian population (Department of Statistics Malaysia, 2021). It is also expected that Malaysians who are diagnosed with dementia will reach 261,000 in the year 2030 according to a report published by the Alzheimer's Disease International in year 2015, and this number can rise with the increase in lifespan (Alzheimer's Disease International et al., 2015). The multiethnicity and its concomitant rich variety of cultural practices among the various races can provide insights into AD with regard to biological and environmental influences.

We conducted an unbiased proteomic analysis using AD patients and normal controls to investigate the disease-associated blood proteome changes in the two major ethnic groups in Malaysia namely the Chinese and Malay. To the best of our knowledge, this study is the first study conducted in Malaysia to compare the blood proteomic profiles between the Chinese and Malay ethnics. The pathways associated with AD and the interactions between the differentiated proteins in these two groups were the main interest of this study. The proteins were profiled using quantitative liquid chromatography-tandem mass spectrometry (LC-MS/MS) based on TMT labelling. Differential expression analysis was performed to identify significant proteins with aberrant abundance. Functional and network analyses were conducted next to identify the protein-protein interactions and pathways of the dysregulated proteins in the two different races. Furthermore, a survey was conducted to compare the sociodemographic characteristics, lifestyle, dietary and behavioural practices of the Chinese and Malay patients. This survey aimed to provide supportive evidence on the possible correlation between external environmental factors to which the two ethnic groups have been subjected and any identified dysregulated proteins. To increase the statistical power and reliability of the findings, a meta-analysis was performed for the identified dysregulated proteins.

## MATERIALS AND METHODS

The proposed pipeline is illustrated in Fig. 1.
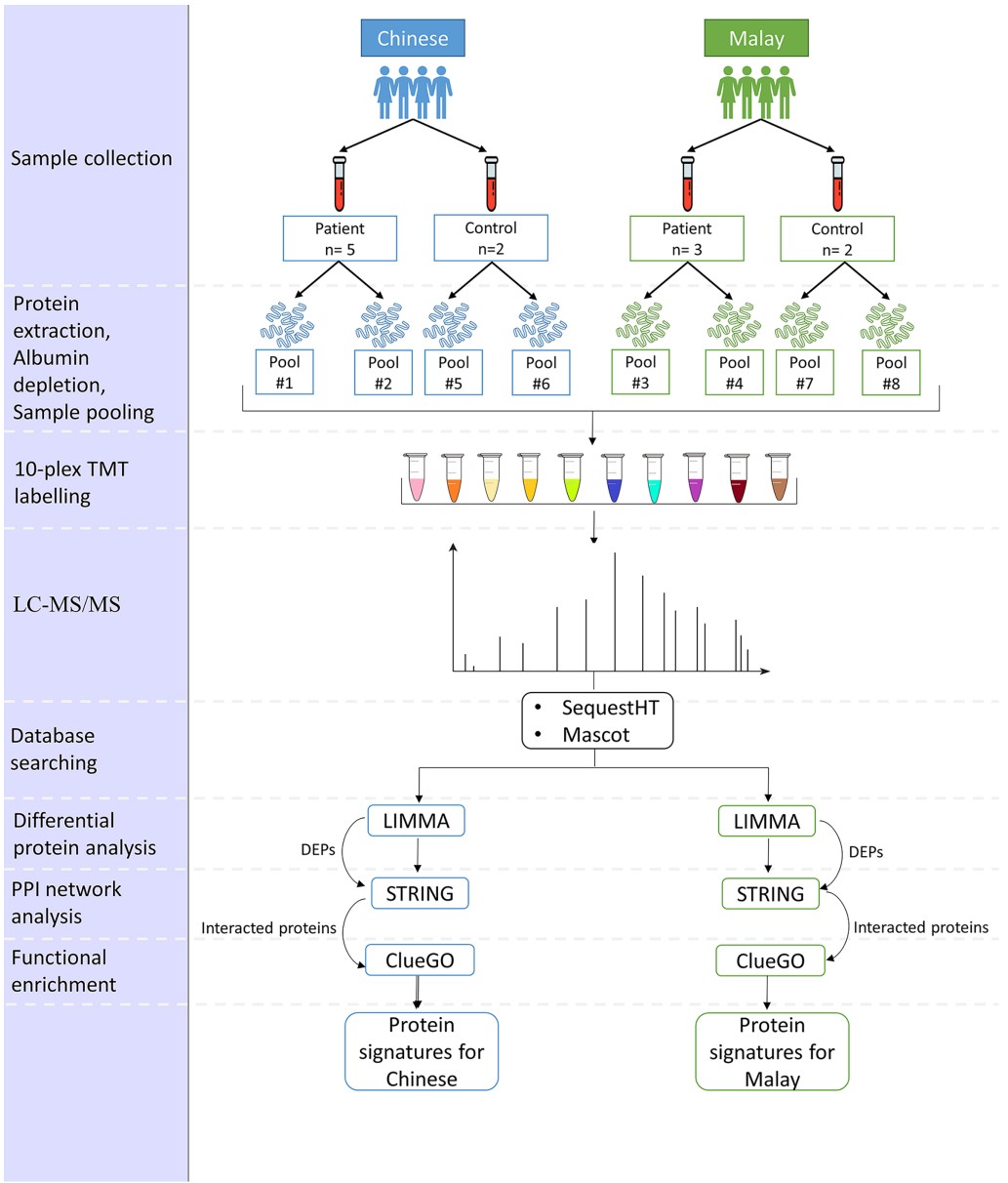

**Figure 1 Pipeline for the comparison of AD-associated blood proteome changes in Chinese and Malay Malaysians.**

## Subjects

A total of 12 subjects, eight patients diagnosed with AD, and four normal controls, were recruited from the Memory and Geriatric clinics at the Universiti Malaya Medical Centre (UMMC), Kuala Lumpur, Malaysia respectively. Written informed consent (BK-MIS-1117-E01) was obtained from all subjects involved in the study. The 12 subjects comprised five AD patients and two normal controls in the Chinese group and three AD patients and two normal controls in the Malay group. All subjects are Malaysian citizens and ≥65 years of age at the time when they were recruited in this study. The selection of AD patients was made based on the criteria stated in Table 1.

**Table 1 Inclusion and exclusion criteria of selecting AD patients and normal controls.**

| Subject | Inclusion criteria | Exclusion criteria |
|---|---|---|
| AD | Malaysian | Under palliative care for other diseases |
| | 65 years old and above | Diagnosed with other type of dementia such as: vascular dementia; lewy body dementia; Parkinson's disease dementia; frontotemporal dementia; creutzfeldt-Jakob disease; wernicke-korsakoff syndrome; normal pressure hydrocephalus; Huntington's disease; down-syndrome dementia. |
| | Shows symptoms of memory deterioration which have worsened with time | |
| | Show symptoms of losing the ability to perform daily function | |
| | Diagnosed with AD for more than 2 months | |
| Normal controls | Malaysian | Patients under palliative care for other diseases |
| | 65 years old and above | Diagnosed with AD/any other type of dementia. |
| | Does not show symptoms of memory deterioration | Diagnosed with a known significant (in the view of the investigator) concurrent medical disease. |
| | Able to perform daily function | |

At the same time, each AD subject was provided with a questionnaire which included sociodemographic details, dietary habits and other behavioural practices. The answers supplied by the subjects were corroborated by the accompanying caregiver.

The above study was approved by the Universiti Malaya Medical Centre (UMMC) Medical Research Ethics Committee with the approval number of 2020114-9193.

## Sample processing

A total of 10 ml of blood was collected from each subject in the BD Vacutainer CAT blood collection tubes and were centrifuged at 1,000 g for 10 min at room temperature. The serum supernatant was collected, centrifuged at 2,500 g for 10 mins at room temperature and then stored at −80 °C until further processing.

## Sample preparation

Depletion of high-abundance proteins was performed on all 12 serum samples using ITSIPREP Albumin Segregation Kit-Solvent (ASKs) (ITSI Bioscience, Johnstown, PA, USA) according to the protocol outlined by the manufacturer. The samples were then pooled according to the race and gender of the subjects, as in Table 2. The protein concentration of the eight pooled samples was determined using the Biorad Bradford Assay Kit (Biorad, Hercules, CA, USA). Sample digestion was carried out according to the manufacturer's instructions. Following digestion, peptides were eluted from the column, dried by vacuum centrifugation and reconstituted in 200 mM HEPES (pH 8.8), with the peptide concentration subjected to Pierce quantitative colourimetric peptide assay (Thermo Scientific, Waltham, MA, USA).

## TMT-labelling and data-dependent acquisition LC-MS/MS

The eight pooled samples and two technical replicates were labelled in a 10-plex TMT label batch as indicated in Table 2. The labelling of the TMT reagent (Thermo Scientific, Waltham, MA, USA) for each sample was according to the Australian Proteome Analysis

**Table 2 List of sample details and the corresponding TMT tags.**

| Sample pool | Label | Patient/control | Race | Gender | Age* | Education background |
|---|---|---|---|---|---|---|
| AD1 | 126 | Patient 1 | Chinese | Female | 75 | Tertiary |
| | | Patient 2 | Chinese | Female | 74 | Secondary |
| | | Patient 3 | Chinese | Female | 75 | No formal |
| AD2 | 127N | Patient 4 | Chinese | Male | 70 | Tertiary |
| | | Patient 5 | Chinese | Male | 75 | Tertiary |
| AD3 | 127C | Patient 6 | Malay | Female | 86 | Tertiary |
| AD4 | 128N | Patient 7 | Malay | Male | 84 | Secondary |
| | | Patient 8 | Malay | Male | 75 | Tertiary |
| AD5@ | 130C | Patient 4 | Chinese | Male | 70 | Tertiary |
| | | Patient 5 | Chinese | Male | 75 | Tertiary |
| Ctrl1 | 128C | Control 1 | Chinese | Female | 84 | Primary |
| Ctrl2 | 129N | Control 2 | Chinese | Male | 73 | Tertiary |
| Ctrl3 | 129C | Control 3 | Malay | Female | 78 | Secondary |
| Ctrl4 | 130N | Control 4 | Malay | Male | 74 | Tertiary |
| Ctrl5@ | 131 | Control 2 | Chinese | Male | 73 | Tertiary |

Notes:
* Age of subjects at the year of recruitment.
@ Technical replicates that are used as standard in controlling variation.

Facility (APAF) SOP MS-096. The pooled peptide mixture was then separated into three fractions using Pierce High pH reverse phase centrifugal columns. HpH fractionated TMT-labelled peptides were subjected to LC-MS/MS analysis. The instrument was operating in positive ion mode, scanning peptide precursors from 350 to 1,850 m/z at 60 k resolution. The ten peptide ions that showed the most intense signals in the survey scan were fragmented by HCD using a normalized collision energy of 33 with a precursor isolation width of 0.8 m/z.

## Protein identification and quantification

The raw data were processed using Proteome Discoverer (Version 2.1.0.81, Thermo Scientific, Waltham, MA, USA). The data were searched using search engines SequestHT and Mascot against a sequence database for the *Homo sapiens*. The parameters for the data processing are shown in Supplemental Methods. The raw quantitative data were used for further bioinformatics analysis.

## Bioinformatics data analysis

A scale normalization was applied and the data were log-2 transformed according to median of all samples before the bioinformatics analysis. Visualization of the protein abundances was carried out in the first step using principal component analysis (PCA) and hierarchical clustering (HC) to explore the patterns of similarity between the two ethnic samples.

Next, two comparison groups were made throughout the analysis: (i) Chinese AD *vs* Chinese Control (CADvC) and (ii) Malay AD *vs* Malay Control (MADvC). Firstly,

differential protein analysis was performed which was followed by HC to identify aberrant proteins as well as to investigate the variation of protein abundances among the two racial groups. Next, protein-protein interaction (PPI) network and functional enrichment analysis were carried out to understand the functional roles and pathways involved by the identified correlated proteins in AD pathogenesis.

## Differential protein analysis

Differential protein analysis was carried out in this study using the Linear Models for Microarray Analysis (LIMMA) package, Bioconductor R, to find out the differentially expressed proteins (DEPs). Proteins with an adjusted $p$-value of <0.05 were selected and ranked as the top DEPs. Benjamini and Hochberg's (BH) method was used as the adjusting method to control the proportion of the false discovery rate within 5% of the total genes. The upregulated and downregulated proteins were grouped according to the marking value of LogFC > 0 and logFC < 0, respectively.

## Hierarchical clustering

Unsupervised HC was performed to identify differentially expressed proteins across the two comparison groups. HC was also carried out using the entire proteomic dataset to observe the expression patterns of proteins from different ethnic groups, as mentioned in the earlier step. Pearson correlation coefficient was applied in this step to obtain the clusters of correlated proteins.

## Functional enrichment and protein-protein interaction network analysis

STRING (version 11.5) (*Szklarczyk et al., 2019*) was used to construct the protein-protein interaction (PPI) networks of the identified DEPs. An interaction score of >0.4 was set to allow protein interactions with medium confidence to be constructed in the networks. Functional enrichment of the interacted proteins identified through PPI networks was computed using the ClueGO application in Cytoscape (*Bindea et al., 2009*). The default background corresponding to the genome-wide genes of *Homo sapiens* was selected. Gene Ontology (GO) terms including biological process, molecular functions and cellular functions were selected. KEGG pathways involved by the interacted proteins were also investigated. A two-sided hypergeometric test with Bonferonni step-down was applied to calculate the $p$-value correction of each term and the network connectivity (Kappa, Turin, Italy) score was set to 0.4. Pathways with $p$-value <0.05 were considered significantly enriched.

## Data analysis on sociodemographic, lifestyle and physical survey of the participants

Statistical analysis was performed on the survey data collected from the 12 subjects to find out the relationship between sociodemographic characteristics, practices of lifestyle, dietary and other behavioural activities with AD among different ethnicities. Scores were given to each question according to their minimum frequency/amount measured daily,
weekly or at one time, depending on their characteristics. Several questions were grouped according to their characteristics and average scores of the group measurement were calculated to reduce the complexity and enhance the normality of the data. In total, 28 variables were used to assess the relationship with the ethnic groups. The nonparametric test of Mann-Whitney U test ($p$-value < 0.05) was used to analyse the continuous variables while Fisher exact test ($p$-value < 0.05) was used for the categorical variables. This is due to the ability of nonparametric methods to tolerate small-size samples and the assumption of data normality is void. The statistical tests were conducted to check whether there are significant differences between the two groups (Chinese and Malay) on the sociodemographic characteristics, practices of lifestyle, dietary and other behavioural activities.

Next, the Pearson correlation coefficient was carried out to find out the correlation of the variables with AD status, in each ethnic group. This test was conducted on the variables that showed significant differences between the Chinese and Malay groups. In this study, a 95% confidence interval was applied, with a $p$-value < 0.05 considered to be significant.

## Meta-analysis

Meta-analysis was performed on the significant DEPs with the relative abundances mean and standard deviation (SD) using Meta-essentials, version 1.5 (*Suurmond, van Rhee & Hak, 2017*). Systematic literature was searched through the database of PubMed based on keywords of "Alzheimer's Disease", "proteomics", "biomarkers" and "human". Studies that reported the mean and SD of the selected dysregulated protein candidates for CADvC (*VDBP, TF, LTF, AHSG, F9, SELENOP, RBP4, ECM1, ITIH1, HGFAC, A1BG, KRT1* and *KRT10*) and MADvC (*APOA4, FGA, C2, C4A* and *ITIH4*) were included. For studies that did not provide relevant information, corresponding authors were contacted through email. Meta-analysis was performed for the DEPs that were reported in at least two studies. The outcomes were presented in the standardized mean differences (SMD) of AD and normal control group, measured using Hedges' algorithm. A confidence interval (CI) of 95% was applied in this study. As variations occurred in the biomarker measurement of different studies (*e.g.*, different proteomic platforms), a random effect model was used. The heterogeneity of the included studies was assessed using Cochran's Q test, t-statistic and $I^2$ index.

# RESULTS

The TMT quantitative proteomic analysis of the serum samples in our study resulted in the identification and quantification of 172 unique proteins. Pre-processing methods were performed on the data, such as the exclusion of the identified highly-abundance serum albumin proteins and proteins with missing values prior to the subsequent downstream analysis. As a result, 163 proteins were included in the analysis.

## Differential protein expression analysis

To observe the cluster conditions of the samples, unsupervised clustering was carried out using PCA and HC. PCA result showed the samples were clustered by ethnicities, as

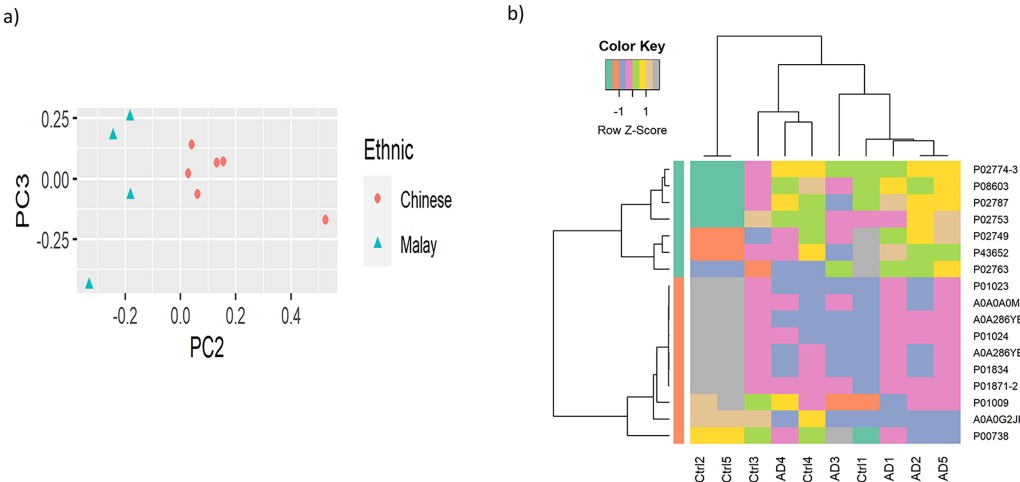

**Figure 2 Visualisation of the samples' clusters using (A) PCA and (B) HC.**

illustrated in Fig. 2A while Fig. 2B shows the HC conducted based on the top 10% of the proteins with the highest variance across the samples. The full version of HC using 163 proteins is attached in Fig. S1. Similarly to the result of PCA, the hierarchical tree clearly indicated the clusters of samples by the two ethnicities. Nevertheless, as the proteins do not perform differential expression analysis yet, the clusters identified are less distinguishable in terms of AD *vs* control.

Differential expression analysis of the quantitative proteomics data ($n = 163$) was performed using LIMMA (adjusted *p*-value < 0.05) across the two comparison groups: CADvC and MADvC. Significant dysregulated proteins identified from each comparison group were listed in Table 3. A Venn diagram of the identified DEPs from the two comparison groups is shown in Fig. 3A. From this finding, it can be observed that most of the DEPs identified in CADvC did not overlap with the DEPs identified in MADvC. Only one protein, F12, was noted in both ethnic groups, and appeared to be upregulated in CADvC but downregulated in MADvC. These suggested that the protein abundances in ethnic groups of Chinese and Malay AD patients might be different.

Figure 3A shows the Venn diagram of the identified DEPs from CADvC and MADvC. HC was again carried out independently with the DEPs identified from each of the respective comparison groups (Figs. 3B and 3C). The resulting heatmaps segregated the AD and control cases in Chinese and Malays, as shown in Figs. 3B and 3C. This verified that the identified DEPs were significantly in distinguishing between AD cases and controls according to the respective race and supported that selected DEPs could be potential markers for AD in Chinese and Malays, respectively.

## PPI network analysis

Following this, PPI was constructed to enhance the understanding of the role and interaction of the correlated dysregulated proteins using the identified DEPs. The PPI networks of correlated proteins identified in CADvC and MADvC (Table 3) are illustrated in Fig. 4.

**Table 3 List of differentially expressed proteins (DEPs) identified through LIMMA in the two comparison groups: CADvC and MADvC.**

**Chinese AD *vs* Chinese control (CADvC)**

| Accession | Protein name | logFC | adj.P.Val | Regulation |
|---|---|---|---|---|
| **Chinese AD *vs* Chinese control (CADvC)** | | | | |
| A0A286YFJ8 | IGHG4 | 0.9111454 | 0.0026895 | Up |
| P00748 | F12 | 0.790919 | 0.0043438 | Up |
| P04264 | KRT1 | 0.773537 | 0.0047613 | Up |
| Q92496 | CFHR4 | 0.9942576 | 0.0072636 | Up |
| P35527 | KRT9 | 0.6347817 | 0.009087 | Up |
| Q04756 | HGFAC | 0.7197487 | 0.0119676 | Up |
| P13645 | KRT10 | 0.7481082 | 0.0155827 | Up |
| P36980 | CFHR2 | 1.0779595 | 0.0222128 | Up |
| P02765 | AHSG | 0.5314173 | 0.0237169 | Up |
| P02753 | RBP4 | 0.9520529 | 0.026608 | Up |
| Q5SQ11 | PTGDS | 0.647988 | 0.0284087 | Up |
| P02774-3 | VDBP | 0.8069054 | 0.0287951 | Up |
| P49908 | SELENOP | 0.5142812 | 0.0298058 | Up |
| P04217 | A1BG | 0.5457707 | 0.0352925 | Up |
| A6NC48 | BST1 | 0.5960251 | 0.0361826 | Up |
| P02787 | TF | 0.6400018 | 0.0372747 | Up |
| P02788 | LTF | 0.7742431 | 0.0396698 | Up |
| Q16610-4 | ECM1 | 0.6953304 | 0.0414427 | Up |
| P00740 | F9 | 0.4752045 | 0.0212655 | Up |
| A0A096LPE2 | SAA2-SAA4 | −0.7740782 | 0.0128821 | Down |
| P02042 | HBD | −1.2352253 | 0.0131224 | Down |
| O75882 | ATRN | −0.5770248 | 0.014957 | Down |
| P19827 | ITIH1 | −0.910063 | 0.0184665 | Down |
| A0A0C4DH31 | IGHV1-18 | −1.6953243 | 0.0283833 | Down |
| A0A087WSY6 | IGKV3D-15 | −1.1552825 | 0.0286905 | Down |
| A0A0G2JMB2 | IGHA2 | −0.8217308 | 0.0420321 | Down |
| **Malay AD *vs* Malay control (MADvC)** | | | | |
| P02671 | FGA | 1.1257808 | 0.0078898 | Up |
| A0A087WSY6 | IGKV3D-15 | 0.5932666 | 0.0413459 | Up |
| A0A0G2JPR0 | C4A | −1.0490485 | 0.0037218 | Down |
| P06681 | C2 | −0.9597209 | 0.0065765 | Down |
| Q12805 | EFEMP1 | −0.8564646 | 0.0090733 | Down |
| Q14624 | ITIH4 | −0.8005078 | 0.0122691 | Down |
| P00748 | F12 | −0.7457226 | 0.015647 | Down |
| A0A286YEY4 | IGHG2 | −0.7778262 | 0.0193615 | Down |
| P06727 | APOA4 | −1.1521863 | 0.0213486 | Down |
| P13646 | KRT13 | −1.1226734 | 0.0283367 | Down |
| P35908 | KRT2 | −0.870911 | 0.0427892 | Down |
| P18065 | IGFBP2 | −0.7772892 | 0.0440745 | Down |
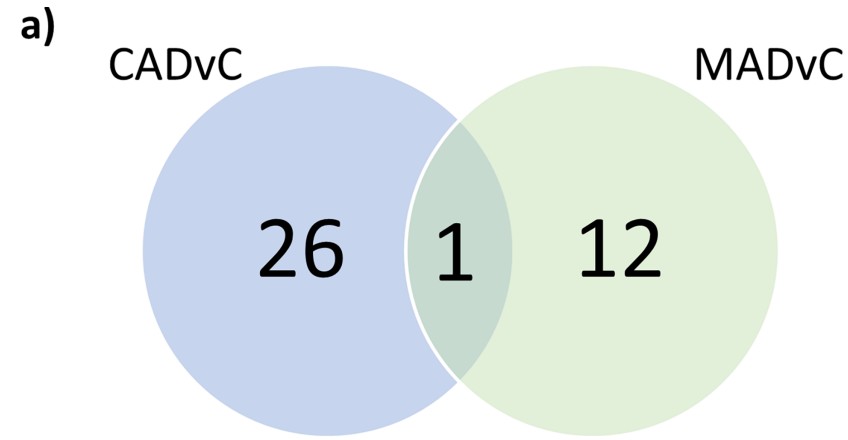

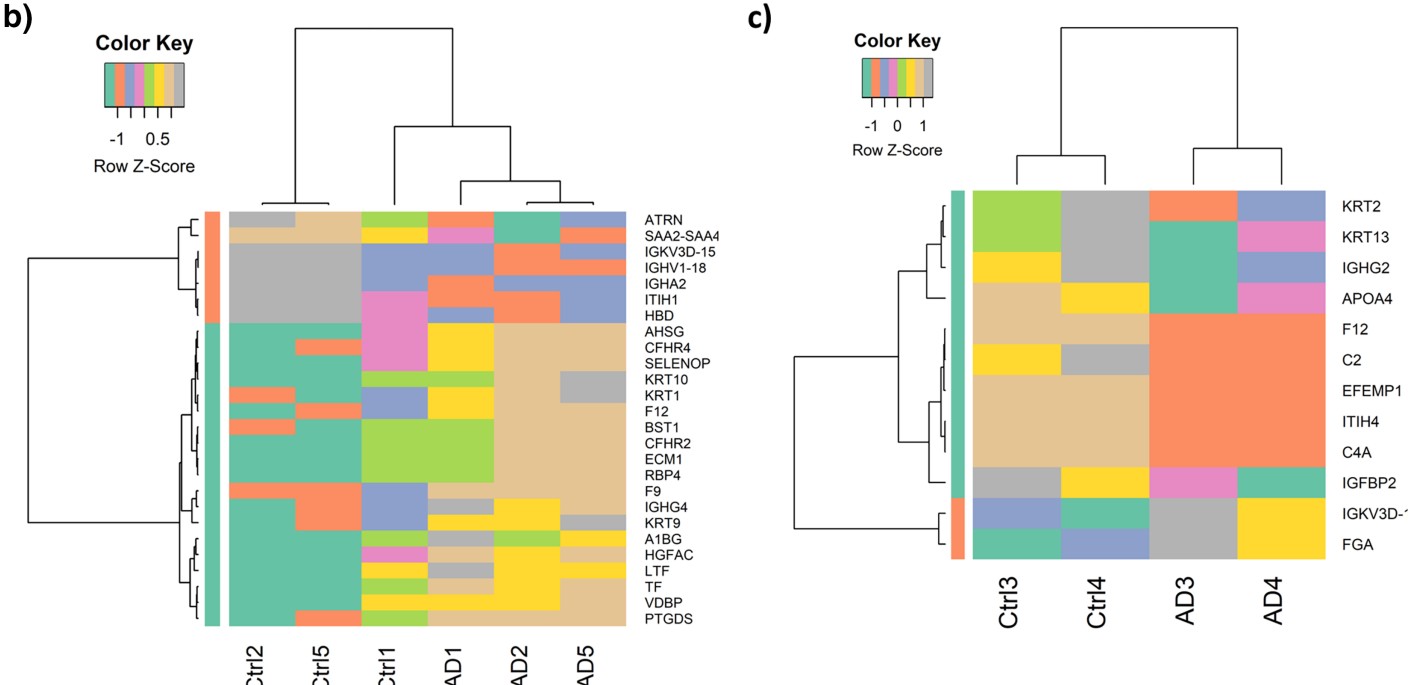

**Figure 3 Differential abundance of proteins observed in AD.** (A) Venn diagram of the identified DEPs from the two comparison groups: CADvC and MADvC. (B) Hierarchical clustering of 27 DEPs obtained from comparison group CADvC using LIMMA. (C) Hierarchical clustering of 10 DEPs obtained from comparison group MADvC using LIMMA.

From the findings illustrated in Fig. 4, it can be seen that all the proteins of immunoglobulin heavy chains were not found in the STRING database. Hence, they were excluded from the PPI networks in this study. This might be due to their nature as part of the polypeptide subunit of an antibody which was not recognised as a complete protein. PPI demonstrated the possibility of one/a group of proteins which affected the expression of other proteins in the regulation *i.e.*, the activation/inhibition of certain pathways. Figure 4A demonstrated the interaction of proteins *AHSG, SELENOP, RBP4, TF, LTF, A1BG, VDBP, ECM1, ITIH1, HGFAC, F9, F12, KRT1, KRT10* in *CADvC*. Proteins *ATRN,*

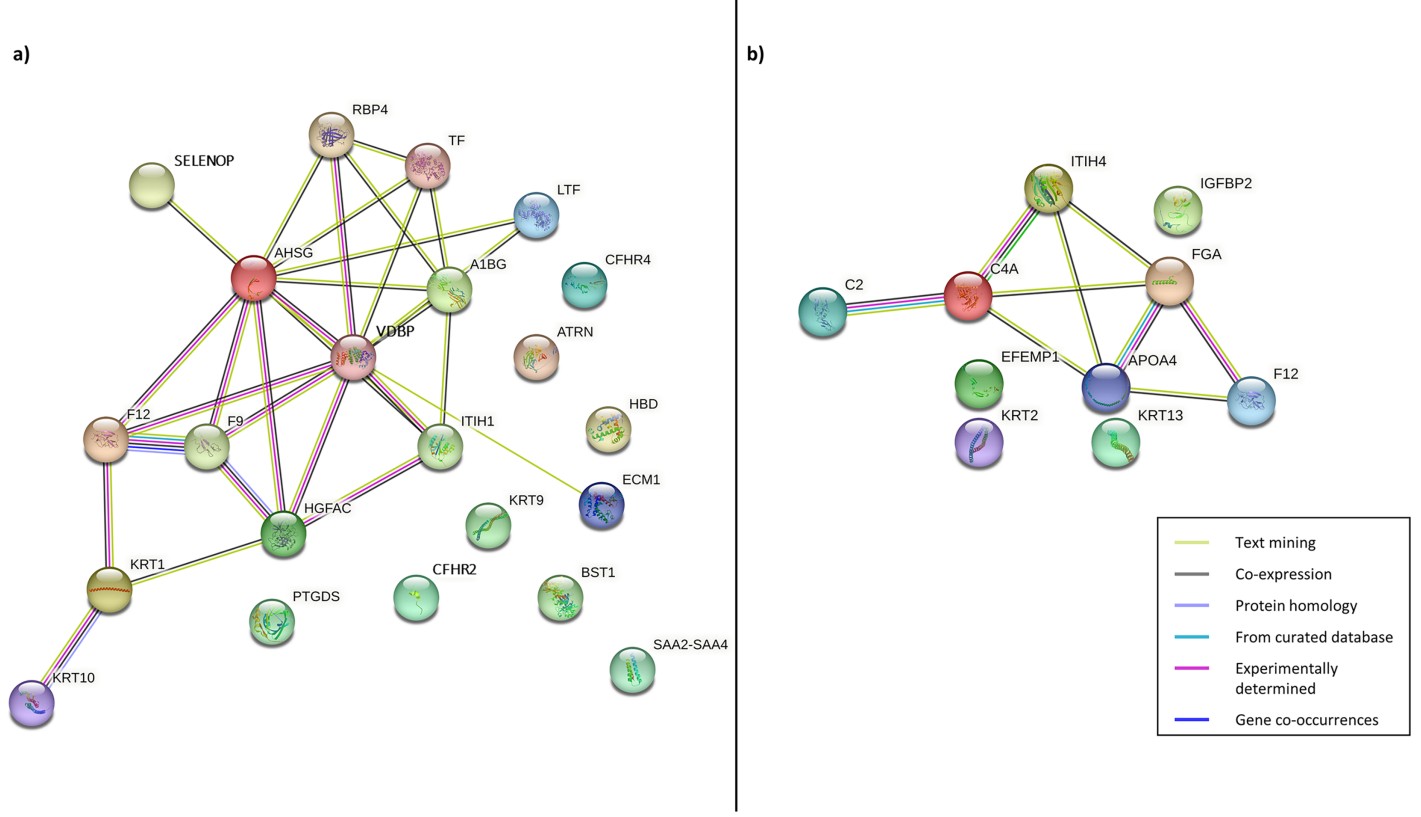

**Figure 4 Protein-protein interaction networks of the correlated proteins identified in (A) CADvC, (B) MADvC.** The proteins are represented by the coloured nodes and are joined by the edges that represent the protein-protein interactions where the joint proteins commonly contribute to some shared functions/roles.

*HBD, CFHR2, CFHR4, KRT9, BST1, PTGDS* and *SAA2-SAA4* did not share similar functions with the other proteins in the network and hence were excluded in the following functional enrichment analysis. In MADvC, proteins *C2, C4A, FGA, ITIH4, APOA4* and *F12* were found to interact with each other, however, the relationship of proteins *EFEMP1, KRT2, KRT13* and *IGFBP2* with the other correlated proteins were uncertain (Fig. 4B).

## Functional enrichment analysis

For further interpretation of the biological functions and pathways of the interacted DEPs, GO enrichment analysis was performed using the ClueGO application in Cytoscape. The resulting GO terms and KEGG pathways which were significantly enriched with the interacted proteins correlated with CADvC and MADvC are shown in Fig. 5. The details of the pathways are included in Table S1. The enriched gene ontology and pathways identified using the interacted proteins in each comparison group were illustrated in Fig. 6.

As demonstrated in Figs. 5 and 6, the dysregulated proteins identified from Chinese and Malay were similarly enriched with the pathways blood microparticle, collagen-containing extracellular matrix and transport, inflammatory response, humoral immune response, defence response and protein metabolic process.

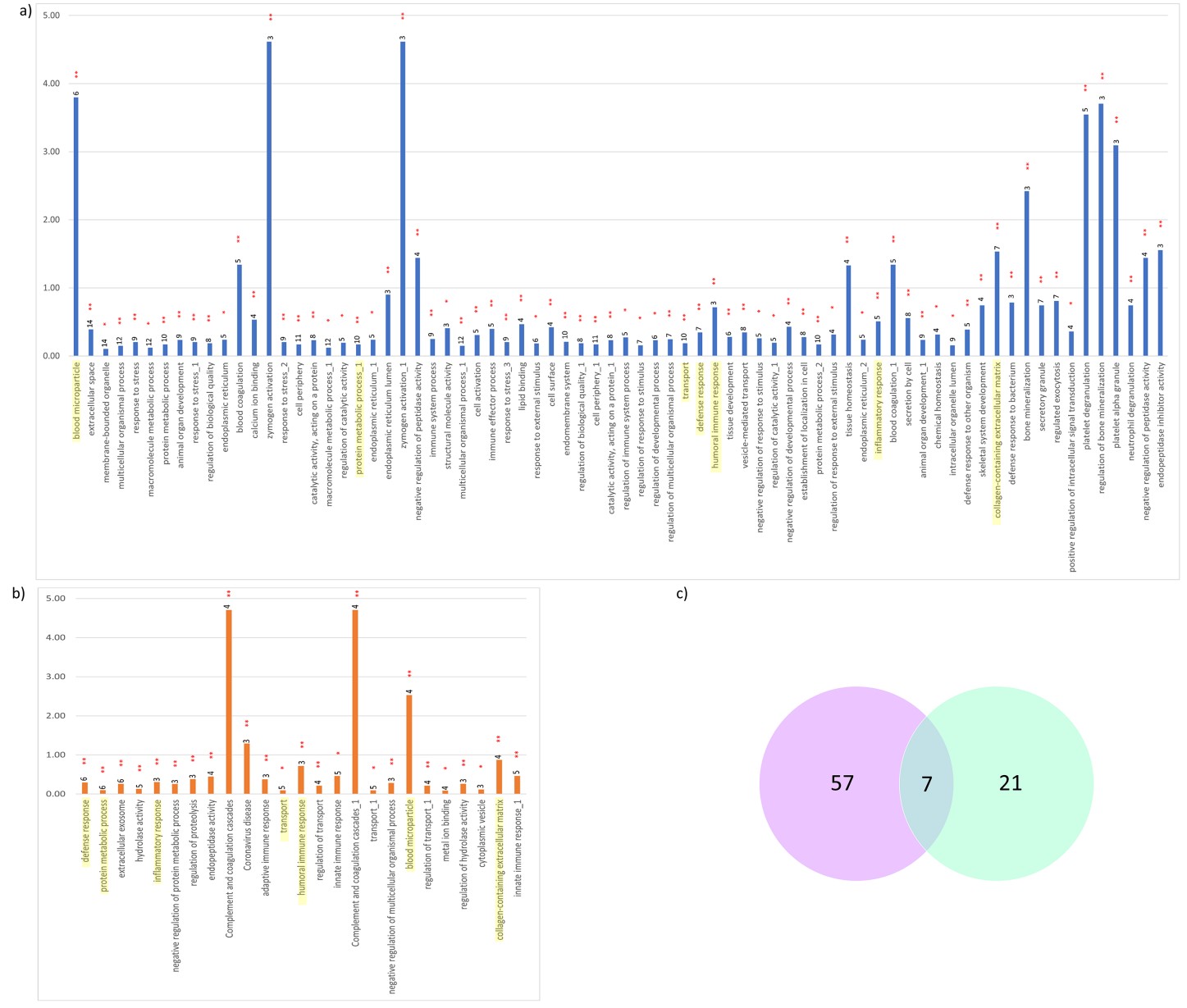

**Figure 5  Significant KEGG pathways and GO terms enriched.** (A) CADvC, (B) MADvC and the number of associated proteins found within each of the pathways. Common pathways between CADvC and MADvC were highlighted in yellow. (C) Venn diagram of the pathways identified.

## Analysis of questionnaire survey data of the subjects by ethnicity

The sociodemographic details and medical conditions of the subjects are shown in Table 4. No significant differences were observed for most of the characteristics between Chinese and Malay ethnic groups. Nevertheless, a significant difference ($p$-value = 0.028) was noted when all the Malay respondents were living with family members while the living situation of the Chinese was distributed to living with spouse, family members, or others such as with maid or old folk's home.

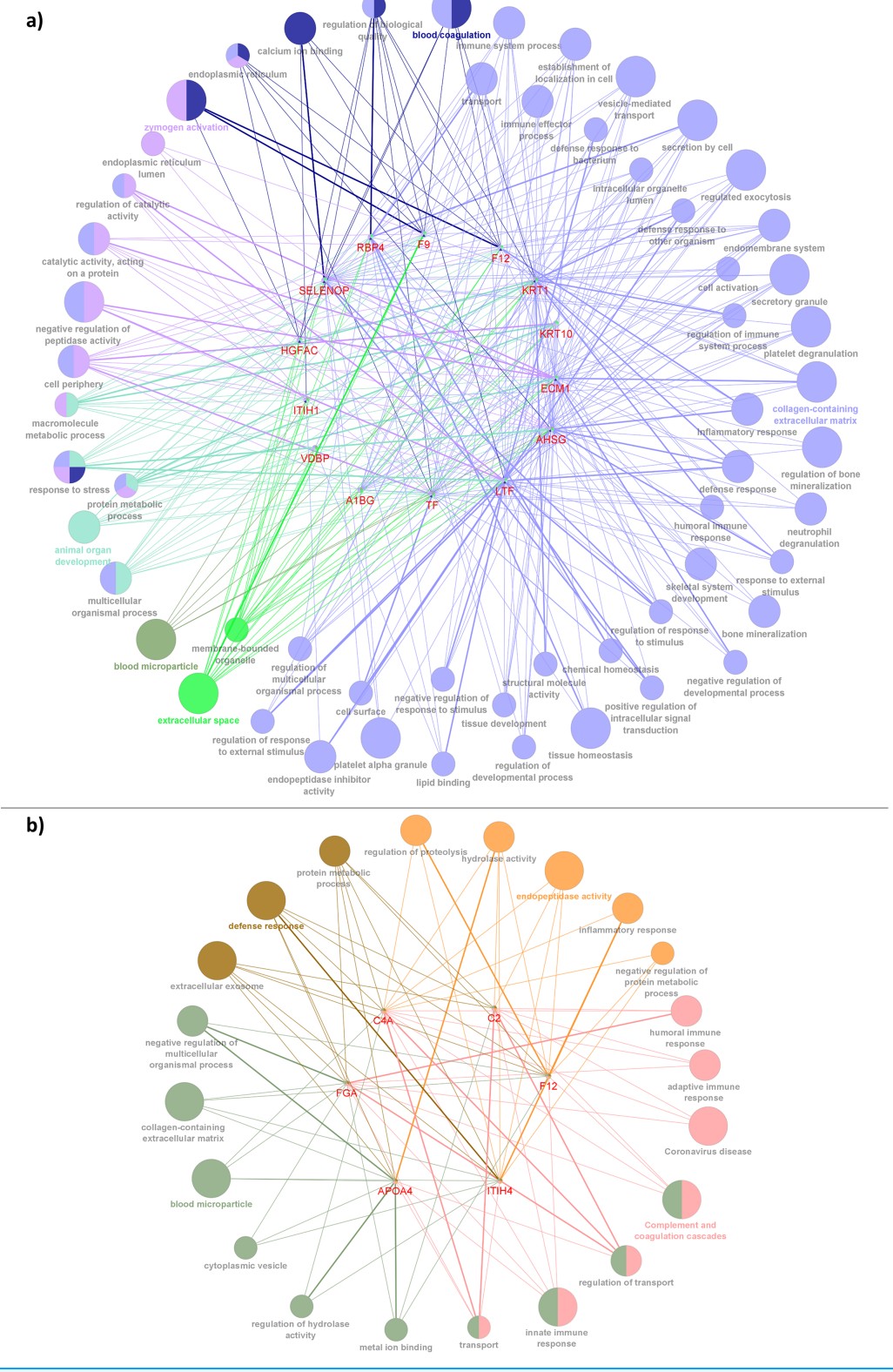

**Figure 6 Network depiction of the identified proteins with enriched GO categories and pathways that related to AD's pathology in (A) CADvC, (B) MADvC.** Each of the nodes represents the enriched pathways and GO terms and the edges indicate the interaction between the proteins and terms.
**Table 4 Sociodemographic and medical details of the subjects.**

| Demographics | | Chinese (*n* = 7) Mean ± SD/*n*(%) | Malay (n = 5) | *p*-value |
|---|---|---|---|---|
| Living area | City centre | 1 (14.3) | 0 | 1 |
| | Town centre | 6 (85.7) | 4 (80) | 1 |
| | Village | 0 | 1 (20) | 0.417 |
| Marital status | Married | 4 (57.1) | 5 (100) | 0.205 |
| | Widow | 3 (42.9) | 0 | 0.205 |
| Living situation | Living with spouse only | 3 (42.9) | 0 | 0.205 |
| | Living with family members (*e.g.*, spouse, siblings and children) | 2 (28.6) | 5 (100) | 0.028 |
| | Others | 2 (28.6) | 0 | 0.47 |
| Highest education | No formal education | 1 (14.3) | 0 | 1 |
| | Primary | 1 (14.3) | 0 | 1 |
| | Secondary | 1 (14.3) | 2 (40) | 0.523 |
| | Polytechnique/university/college | 4 (57.1) | 3 (60) | 1 |
| Height | | 159.29 ± 5.77 | 160.28 ± 4.65 | 0.744 |
| Weight | | 54.14 ± 7.16 | 57.24 ± 8.15 | 0.53 |
| Systolic blood pressure | | 115.86 ± 5.15 | 126.8 ± 21.39 | 0.678 |
| Family history of AD | | 2 (28.6) | 0 | 0.47 |
| Medical conditions/history | Diabetes mellitus | 2 (28.6) | 1 (20) | 1 |
| | Hypertension | 2 (28.6) | 2 (40) | 1 |
| | Visual disease | 3 (42.9) | 2 (40) | 1 |
| | Heart disease | 0 | 2 (40) | 0.152 |
| | Asthma | 1 (14.3) | 1 (20) | 1 |
| | Hepatitis | 2 (28.6) | 1 (20) | 1 |
| | Arthritis | 1 (14.3) | 1 (20) | 1 |
| | Others | 1 (14.3) | 2 (40) | 0.523 |
| History of head injury | | 1 (14.3) | 0 | 1 |
| Employment status | Unemployed | 1 (14.3) | 0 | 1 |
| | Self-employed | 1 (14.3) | 0 | 1 |
| | Retired | 5 (71.4) | 5 (100) | 1 |

Following the statistical analysis as shown in Table 5, the parameters surveyed in the questionnaire did not show a significant difference between Chinese and Malay subjects, except for the consumption of sweet drinks/foods; the sugar consumption of Malay patients (mean of 8.29 ± 1.05) was significantly higher compared to their Chinese counterparts (mean of 4.03 ± 1.83), with a *p*-value of 0.023.

**Table 5 Lifestyle and physical practices of participants by ethnicity.**

| Practices | | Chinese (*n* = 7) Mean ± SD/*n*(%) | Malay (*n* = 5) | *p*-value |
|---|---|---|---|---|
| Lifestyle and dietary practices | | | | |
| Duration of sleep (hours/night) | | 7.9 ± 0.22 | 7.33 ± 0.76 | 0.723 |
| No. of times wake up at night (times/night) | | 1.2 ± 0.1 | 2 ± 0 | 0.259 |
| Smoker | Smoker | 1 (14.29) | 3 (60) | 0.222 |
| | Never | 6 (85.71) | 2 (40) | 0.222 |
| Cigarettes (cigarettes/week) | | 20 ± 52.92 | 63 ± 62.61 | 0.208 |
| No. of times skip meals (breakfast, lunch or dinner) (times/week) | | 0.19 ± 0.5 | 0 | 0.5 |
| Eating style | Vegetarian | 1 (14.29) | 0 | 1 |
| | Non-vegetarian | 6 (85.71) | 5 (100) | 1 |
| Fast food/fried food/food high in salt and fat (times/week) | | 1 ± 0 | 2.4 ± 1.92 | 0.103 |
| Sweet drinks/food (*e.g.*, snacks, soda, dessert, fruits) (times/week) | | 4.03 ± 1.83 | 8.29 ± 1.05 | 0.023 |
| Alcohol consumption (unit/occasion) | | 0.6 ± 1.34 | 0 | 0.259 |
| Physical practices | | | | |
| Mode of transportation | Private transport | 7 (100) | 5 (100) | – |
| Time spent on transportation (mins/time) | | 12.86 ± 23.6 | 12 ± 16.43 | 0.922 |
| No. of days of going out (days/week) | | 2.29 ± 1.8 | 2.6 ± 2.88 | 1 |
| Types of outdoor activities | Dining | 3 (42.86) | 3 (60) | 1 |
| | Exercise | 1 (14.29) | 1(20) | 1 |
| | House visit | 1 (14.29) | 1 (20) | 1 |
| | Entertainment | 1 (14.29) | 1 (20) | 1 |
| | Run errands | 2 (28.57) | 1 (20) | 1 |
| | Medical appointment | 7 (100) | 5 (100) | – |
| Level of exercise | No exercise | 1 (14.29) | 0 | 1 |
| | Light exercise | 4 (57.14) | 5 (100) | 0.205 |
| | Moderate exercise | 2 (28.57) | 0 | 0.47 |
| Time spent on exercise (mins/day) | | 14.28 ± 9.32 | 20 ± 7.91 | 0.283 |
| Sitting (mins/day) | | 124.29 ± 61.06 | 174 ± 13.42 | 0.17 |

**Table 6 Pearson correlation coefficient test on the parameters of living with family members and consumption of sweet drink/food in Chinese and Malay ethnic groups.**

| Parameters | Pearson correlation coefficient (r) (*p*-value) | |
|---|---|---|
| | Chinese | Malays |
| Living with family members (*e.g.*, spouse, siblings and children) | 0.4 (0.374) | – |
| Sweet drinks/food (*e.g.*, snacks, soda, dessert, fruits) | −0.551 (0.2) | 0.879 (0.049) |

From Tables 4 and 5, two parameters showed significant differences between Chinese and Malay participants which are living with family members and consumption of sweet drinks/food. Next, the Pearson correlation coefficient was performed on these two

parameters to find the correlation of the parameters with the Chinese and Malay groups. The correlation results are shown in Table 6. Low correlations were found for both parameters (r = 0.4 and r = −0.551 respectively) with the Chinese ethnic. As for the Malay group, all the Malay participants lived with their family members (*e.g.*, spouse, siblings and children), hence no deviation (SD = 0) and no correlation were calculated for this parameter. On the contrary, the consumption of sweet drinks/food showed a significantly high correlation (r = 0.879, *p*-value = 0.049) with the Malay group.

## Meta-analysis

From the PubMed literature search, 23 articles were identified to be related and were accessed for suitability. From there, 13 articles were included in the systematic review and a total of eight studies including the outcomes from the current study were included in the meta-analysis. Tables S2 and S3 presented the forest plots of the dysregulated proteins identified in CADvC and MADvc respectively, among the AD and normal control subjects. For the meta-analysis of the dysregulated proteins identified from CADvC (Table S2), protein *HGFAC* was excluded from the analysis as no relevant information could be found. Meanwhile, proteins *TF* (SMD= −0.18, 95% CI= [−1.09 to 0.73], *p* = 0.01, r = 0.09), *AHSG* (SMD= −0.24, 95% CI = [−1.12 to 0.64], *p* < 0.001, r = 0.13) and *A1BG* (SMD = −0.21, 95% CI = [−1.89 to 1.47], *p* = 0.01, r = 0.11) showed significant down-regulation in AD compared to control. Notably, all of these proteins showed weak dysregulations with low effect size (r < 0.3) in the data. From the outcomes presented by the meta-analysis of the dysregulated proteins identified in MADvC (Table S3), proteins *APOA4* (SMD = 0.29, 95% CI = [−0.97 to 1.55], *p* < 0.001, r = 0.14) and *C4A* (SMD = 0.24, 95% CI = [−20.59 to 21.06], *p* < 0.001, r = 0.39) were found to be significantly upregulated in AD compared to normal control. In this comparison group, protein *APOA4* showed weak dysregulations while protein *C4A* showed medium dysregulation (moderate effect size, r < 0.5) among the AD group.

## DISCUSSION

This study aimed to provide new insight into Alzheimer's disease *via* a study of proteomics using sampling from the two major races from the multi-ethnic population in Malaysia. It is presumed that the genetic constitution of different ethnic groups differs, thus, the gene regulation product, proteins, might on this account alone already vary (*Galanter et al., 2017*; *Huang, Shu & Cai, 2015*). However, we also felt that by looking at the proteins, rather than being confined to genomes, could at the same time encompass environmental influencers in the development of AD.

In this study, the number of samples from the Chinese (seven samples) and Malays (five samples), do not reflect the general composition of our population, in which the Chinese form 22.6% and the Malays form 69.6% of the population (*Department of Statistics Malaysia, 2021*). The lower number of Malays studied was due to the attendance pattern at the Memory Clinic where ~50% of the patients who visited the clinic were Chinese while only ~16% of the patients were Malays. To minimise the potential confounding related to medications and comorbidities, patients who were under palliative care for other diseases

were excluded from the recruitment of the subjects in this study. Although the sample quality of AD proteomics is unavoidably affected by the cofounding factors, literature has shown that the confounding effects are relatively smaller on the analysis of whole proteome compared to that on the modified proteome (*i.e.*, ubiquitinome and phosphatome) due to the characteristic of modified proteins which are highly unstable and dynamic (*Bai et al., 2021*). In the current proteomic study, the pooling strategy applied to the samples by grouping subjects with similar gender, ethnicity and disease condition together averages the confounding effects and at the same time increases the statistical power (*Johnstone et al., 2012*). Also, as shown from the result of the survey analysis of the demographic data, none of the possible comorbidity factors showed a significant correlation with AD (Tables 5 and 6), other than the practice of sweet intakes among the Malay population.

The results from differential expression analysis comparing CADvC and MADvC (Table 3 and Fig. 3) indicated that DEPs differed between the two ethnic groups. It suggested that different protein regulations were involved in the AD development of Chinese and Malay patients. Among the DEPs identified, Coagulation factor XII (*F12*) is the only protein that is commonly found among the CADvC and MADvC. There is evidence that the depletion of *F12* in mice lessens fibrin deposition and lessens cognitive decline (*Chen et al., 2017*; *Park et al., 2021*). The activation of *F12* triggers thrombosis in plasma which in turn induces inflammation in the AD pathogenesis pathway (*Singh et al., 2021*). In this study, *F12* was found to show upregulated activity in CADvC but downregulated in MADvC. This shows that the roles of this protein on AD development might vary in different ethnic groups, due to the effect of different environmental factors and lifestyle practices. A study was conducted to test the average activity of *F12* in the Chinese population (*Han et al., 2015*). Mutations were found in subjects who were detected with abnormal low activities of *F12* which suggested the potential of *F12* dysregulation in correlating with the Chinese population. However, our study did not support this finding and further clarification is needed to confirm this.

Despite the divergence in proteins identified between CADvC and MADvC (Figs. 5 and 6), several pathways were commonly found between the groups. Other than blood microparticles, collagen-containing extracellular matrix and transport are pathways relating to the cellular and functional component of the blood, the other pathways are found to exhibit a close connection with AD pathogenesis. As such, the cellular and molecular changes in inflammatory, humoral immune and defence responses are found to be linked with neuroinflammation, which is one of the critical factors contributing to AD pathogenesis. Inflammation response occurring in the brain is a well-established core feature for AD development (*Kinney et al., 2018*). On top of that, there are increasing evidences demonstrating the relationship between peripheral inflammation with cognitive dysfunction in AD (*Leung et al., 2013*; *Park, Han & Mook-Jung, 2020*). Inflammatory signals have been shown in the blood peripheral system and act as a communication route to the cytokines levels in brain (*Leung et al., 2013*; *Park, Han & Mook-Jung, 2020*). Humoral immune response, where defence response is part of the system, refers to the initiation of antibody molecule production in blood. In AD, circulating anti-*Aβ* antibodies

such as B cells and immunoglobulins enhance neuroprotection over the development of *Aβ*, hence attenuating AD pathogenesis potentially (*Cao & Zheng, 2018*; *Montero-Calle et al., 2020*). The dysfunction in protein metabolism can cause neurodegenerative diseases, including AD. One of the most remarkable examples is the dysregulation in the *Aβ* mechanism, where the failure can lead to the accumulation of plaques in the brain (*Kaddurah-Daouk et al., 2013*; *Montero-Calle et al., 2020*; *Yan et al., 2020*).

In CADvC, 14 proteins showed an association with each other (Fig. 4A). These proteins showed consistent enrichment with extracellular matrix (ECM), as shown in Fig. 6A. This observation implicated the prominence of ECM in the causal pathogenesis of AD. The dysregulation of ECM has been found to be related to the development of AD in several ways, including the triggering/inhibition of Aβ aggregation, interaction with tau protein, relieving oxidative stress and reducing inflammatory response (*Sun et al., 2021*). This can be further corroborated by the role of the identified proteins, as shown in Table S4.

Recent studies found that the consumption of alcohol is relevant to the inflammatory mechanism in inducing AD by reducing Aβ uptake by the primary microglia (*Eid, Mhatre & Richardson, 2019*; *Heymann et al., 2016*; *Kalinin et al., 2018*; *Langballe et al., 2015*). According to the survey analysis done in this study, the habit of alcohol consumption, although it was not significant, was higher among the Chinese group. Alcohol consumption is not practised among the Malay population due to their religious belief of Islam. In accordance with these findings, this study highlighted the involvement of the dysregulated proteins identified among the Chinese group in the pathways related to an inflammatory response. Reports showed that alcohol drinkers tend to have more severe cognitive decline compared to those infrequent or non-drinkers (*Eid, Mhatre & Richardson, 2019*; *Heymann et al., 2016*; *Langballe et al., 2015*).

Notably, the roles of some identified proteins (*e.g.*, *ITIH1, HGFAC, A1BG, KRT1*, and *KRT10)* in AD development are yet to be discovered. This study uncovered a group of previously unknown proteins that are associated with each other, suggesting the potential for their co-regulation on ECM-related pathways and their roles in AD pathogenesis. Hence, they might serve as potential protein markers which require attention in further research.

Meta-analysis was conducted using the set proteins identified in CADvC in order to increase the statistical power and reliability of the result of this study. The aberrant levels of proteins *TF, AHSG* and *A1BG* among AD were confirmed through the findings of the meta-analysis.

The proteins identified from the MADvC support the influence of neuroinflammation in the pathogenesis of AD. The findings of the roles of these proteins in leading to neuroinflammation further enhanced the contribution of functional terms defence response, protein metabolic process and ECM in AD pathophysiology relating to Malays. The roles of the proteins and their relationship with AD are summarized in Table S5. The reliability of the roles and functions of proteins *APOA4* and *C4A* in AD was supported by the findings of the meta-analysis conducted in this study which showed significant upregulation activities among the AD group when compared to the normal controls.

Interestingly, the expressions of a few identified proteins were found to be influenced by the living environment factors. The association of the *VDBP* with ethnicity was found to be strong in several studies related to vitamin D deficiency in Malaysia (*Chin et al., 2014*; *Rahman et al., 2004*; *Shafinaz & Moy, 2016*). Protein *VDBP* functions to maintain the levels of vitamin D in plasma by promoting its reabsorption in the kidney (*Bouillon et al., 2020*; *Rozmus et al., 2020*). Previous studies indicated the association of vitamin D with Aβ clearance in the brain, attenuating neurodegeneration (*Bivona et al., 2019*, *2021*). The consumption of nutritional supplements (including vitamin D) in the Chinese population was found to be higher compared to the other ethnicities in Malaysia (*Abdullah, Teo & Foo, 2016*; *Saffian et al., 2021*). Furthermore, sunlight exposure is the main source of vitamin D in Malaysia. The lower level of melanin content in the skin and the common clothing practices encourage higher vitamin D absorption from the sunlight in the Malaysian Chinese group (*Saffian et al., 2021*; *Shafinaz & Moy, 2016*). Hence, the higher vitamin D prevalence among the Chinese population might explain the high abundance of protein *VDBP* found among Chinese AD patients (*Woon et al., 2019*).

*APOA4* is found to be involved in the regulation of blood sugar levels by improving insulin sensitivity and promoting sugar uptake (*Li et al., 2017*). Studies have found the expression of *APOA4* to be affected by the sugar levels in the brain (*Liu et al., 2004*; *Wang et al., 2015*). In this study, *APOA4* showed significantly differentiated abundance with AD in the Malay group (MADvC). The presence of *APOA4* was found to lead to the deterioration of cognitive competency in healthy Malay individuals (*Abu Bakar et al., 2019*). In addition, the relationship of *APOA4* with sugar intake in Malays is further supported by the data analysis from our survey, where there was significantly higher sugar consumption in the Malays compared to the Chinese. Numerous studies have also demonstrated marked differences in sugar intake habits across different ethnic groups, where Malays were found to have a higher affinity toward food and drinks with higher sugar levels (*Cheah et al., 2019*; *Eng et al., 2022*; *Zainuddin et al., 2018*).

One limitation of this study is the small sample size collected. The heterogeneity of dementia (*i.e.*, vascular dementia, Lewy Bodies dementia, AD, *etc.*) with different clinical and progress presentations in patients who visited the clinics resulted in the restriction of the number of patients that could be recruited into this study. Furthermore, as blood sampling is an invasive sampling method, some of the patients refused to provide consent. As a result, recruitment was limited restricting the proteomic analysis. Limited sample size in the proteomic study can lead to biased results and conclusions and to overcome this, the multiplex strategy of pooling samples was applied in this study. The samples were pooled according to gender and disease condition of the respective ethnicity of the subjects so that similar proteins from different samples but the same ethnicity could be eluted at the same mass-to-charge ratio (m/z) and at the same time. By this, the relative abundances of different proteins could be directly compared and the problem of the bias can be reduced (*Arul & Robinson, 2018*).

In this study, the differential protein analysis was carried out using limma with proteins of adjusted *p*-value $< 0.05$ considered significant aberrant. Notably, $\log_2 FC$ was not applied as a filtering criterion in selecting differentiated proteins but used as a marking value in

evaluating the regulation of the dysregulated proteins. Imposing fold change on a small sample size experiment often causes uncertainty in the sample variability estimation (*Kammers et al., 2015*). The application of a commonly used fold change threshold (*i.e.*, 1.5) could also result in the loss of information or unreproducible outcomes (*Schwammle, León & Jensen, 2013*). Besides that, to reduce the compulsion imposed by the small sample size of this experiment, moderated t-statistics rather than the ordinary t-test was computed using the empirical Bayes method in limma. The statistical principles implemented in limma help to reduce the problems inherited in small experiments. The statistical method of empirical Bayes allows the "borrowing" of strength between the proteins in a dynamic way, reducing the number of false positive and false negative rates (*Kammers et al., 2015*; *Ritchie et al., 2015*). Hence, the statistical power and reliability of the results were improved for experiments with a small number of replicates, as contained in this study (*Ritchie et al., 2015*).

This study suggests the preliminary proteomic profiles of AD in Chinese and Malay groups in Malaysia. Although the findings of this study are consistent with the literature, they nonetheless represent the dysregulation of proteins found within a small sample size dataset and will require further validation. First, the expression of the selected biomarkers from this study could be confirmed through *in vitro* experiments such as absolute protein quantification. Next, future research should also expand towards the deep evaluation of the described mechanisms/networks identified in this study such as using bioimaging techniques to effectively improve the chances of reducing neuroinflammation/ metabolic dysregulation that leads to AD pathogenesis. Thirdly, the prospects should include more subjects in future studies and expand to the other ethnicities in Malaysia. The multi-cultural genetic constitution could provide a better understanding of AD pathogenesis by bridging the genetic, molecular and network factors to different behavioural/environmental factors. Furthermore, the proposed method could be expanded into multi-omics analysis to include more data from the other omics (*i.e.*, transcriptomics and metabolomics) so that the characteristics, functions and mechanisms from the other omics could be linked and connected. Further investigations may provide an opportunity to evaluate the understanding of AD pathogenesis and different physiological mechanisms together with the physical environment in designing new diagnostic approaches/therapies.

## CONCLUSIONS

The expansion of the ageing population has indirectly led to the increment of AD cases reported. With regard to that, the study of the interaction between the dysregulated proteins may provide an understanding of the characterization of AD pathophysiology. The variation of protein abundances in different ethnic populations that are possibly affected by different behavioural practices caught more attention in the research relating to AD study. This study proposes an integrated method that combines proteomic profiling and differential expression analysis between AD patients and normal controls for Chinese and Malays in Malaysia. The pathways and protein-protein interactions of the identified protein signatures were analyzed using functional enrichment analysis. The results showed that the dysregulated proteins identified from Chinese samples were significantly enriched

to the pathway leading to Aβ/tau protein, oxidative stress and inflammation and the dysregulated proteins for Malay groups were more related to neuroinflammation. The results were supported by previous studies on the developmental mechanisms of AD in Malaysia, especially among the Chinese and Malay populations. The survey conducted further interpreted the impact of different lifestyle practices on the variation of dysregulated protein identified in this study. The significant difference in sugar consumption practices between Chinese and Malay groups supports the relation of dysregulated *APOA4* protein in the Malay groups. An additional meta-analysis conducted further supported the significant aberrances in levels of proteins *TF, AHSG, A1BG, APOA4* and *C4A* among AD patients compared to the normal controls. This study suggests preliminary findings on AD-related proteomics analysis and its relation to the environmental factors in Chinese and Malay populations in Malaysia. This effort should be expanded to include more subjects and ethnicities to have a better understanding of the pathogenesis of AD in Malaysia.

### Funding
This work was supported by the UM International Collaboration Grant, project number ST041-2022 and the Fundamental Research Grant Scheme (FRGS), Ministry of Higher Education Malaysia, project number FRGS/1/2019/SKK06/UM/02/5. The funders had no role in study design, data collection and analysis, decision to publish, or preparation of the manuscript.

### Grant Disclosures
The following grant information was disclosed by the authors:
UM International Collaboration Grant: ST041-2022.
Fundamental Research Grant Scheme (FRGS), Ministry of Higher Education Malaysia: FRGS/1/2019/SKK06/UM/02/5.

### Competing Interests
The authors declare that they have no competing interests.

### Author Contributions
- Mei Sze Tan conceived and designed the experiments, performed the experiments, analyzed the data, prepared figures and/or tables, authored or reviewed drafts of the article, and approved the final draft.
- Phaik-Leng Cheah conceived and designed the experiments, authored or reviewed drafts of the article, and approved the final draft.
- Ai-Vyrn Chin conceived and designed the experiments, authored or reviewed drafts of the article, and approved the final draft.
- Lai-Meng Looi conceived and designed the experiments, authored or reviewed drafts of the article, and approved the final draft.

- Siow-Wee Chang conceived and designed the experiments, analyzed the data, authored or reviewed drafts of the article, and approved the final draft.

## Human Ethics

The following information was supplied relating to ethical approvals (*i.e.*, approving body and any reference numbers):

This study was approved by the Universiti Malaya Medical Centre (UMMC) Medical Research Ethics Committee (2020114-9193).

## Data Availability

The mass spectrometry proteomics data are available at the ProteomeXchange Consortium *via* the PRIDE (*Perez-Riverol et al., 2022*) partner repository: PXD040566.

The raw data for the survey analysis is available in the Supplemental File.

## Supplemental Information

Supplemental information for this article can be found online at http://dx.doi.org/10.7717/peerj.17643#supplemental-information.

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

## FURTHER READING

**Ashraf A, Ashton NJ, Chatterjee P, Goozee K, Shen K, Fripp J, Ames D, Rowe C, Masters CL, Villemagne V. 2020.** Plasma transferrin and hemopexin are associated with altered Aβ uptake and cognitive decline in Alzheimer's disease pathology. *Alzheimer's Research & Therapy* **12(1)**:1–13 DOI 10.1186/s13195-020-00634-1.

**Begic E, Hadzidedic S, Obradovic S, Begic Z, Causevic M. 2020.** Increased levels of coagulation factor XI in plasma are related to Alzheimer's disease diagnosis. *Journal of Alzheimer's Disease* **77(1)**:1–12 DOI 10.3233/JAD-200358.

**Bennett S, Grant M, Creese AJ, Mangialasche F, Cecchetti R, Cooper HJ, Mecocci P, Aldred S. 2012.** Plasma levels of complement 4a protein are increased in Alzheimer's disease. *Alzheimer Disease & Associated Disorders* **26(4)**:329–334 DOI 10.1097/WAD.0b013e318239dcbd.

**Bermejo-Pareja F, Del Ser T, Valentí M, de la Fuente M, Bartolome F, Carro E. 2020.** Salivary lactoferrin as biomarker for Alzheimer's disease: brain-immunity interactions. *Alzheimer's & Dementia* **16(8)**:1196–1204 DOI 10.1002/alz.12107.

**Bian Z, Yamashita T, Shi X, Feng T, Yu H, Hu X, Hu X, Bian Y, Sun H, Tadokoro K. 2021.** Accelerated accumulation of fibrinogen peptide chains with Aβ deposition in Alzheimer's disease (AD) mice and human AD brains. *Brain Research* **1767(34)**:147569 DOI 10.1016/j.brainres.2021.147569.

**Bishnoi RJ, Palmer RF, Royall DR. 2015.** Vitamin D binding protein as a serum biomarker of Alzheimer's disease. *Journal of Alzheimer's Disease* **43(1)**:37–45 DOI 10.3233/JAD-140042.

**Chen M, Xia W. 2020.** Proteomic profiling of plasma and brain tissue from Alzheimer's disease patients reveals candidate network of plasma biomarkers. *Journal of Alzheimer's Disease* **76(1)**:349–368 DOI 10.3233/JAD-200110.

**Cui Y, Huang M, He Y, Zhang S, Luo Y. 2011.** Genetic ablation of apolipoprotein A-IV accelerates Alzheimer's disease pathogenesis in a mouse model. *The American Journal of Pathology* **178(3)**:1298–1308 DOI 10.1016/j.ajpath.2010.11.057.

**De Luca C, Virtuoso A, Maggio N, Papa M. 2017.** Neuro-coagulopathy: blood coagulation factors in central nervous system diseases. *International Journal of Molecular Sciences* **18(10)**:2128 DOI 10.3390/ijms18102128.

**Du X, Qiu S, Wang Z, Wang R, Wang C, Tian J, Liu Q. 2014a.** Direct interaction between selenoprotein P and tubulin. *International Journal of Molecular Sciences* **15(6)**:10199–10214 DOI 10.3390/ijms150610199.

**Du X, Zheng Y, Wang Z, Chen Y, Zhou R, Song G, Ni J, Liu Q. 2014b.** Inhibitory act of selenoprotein P on Cu+/Cu2+-induced tau aggregation and neurotoxicity. *Inorganic Chemistry* **53(20)**:11221–11230 DOI 10.1021/ic501788v.

**Eldem E, Barve A, Sallin O, Foucras S, Annoni J-M, Schmid AW, Alberi Auber L. 2022.** Salivary proteomics identifies transthyretin as a biomarker of early dementia conversion. *Journal of Alzheimer's Disease Reports* **6(1)**:31–41 DOI 10.3233/ADR-210056.

**Festoff BW, Citron BA. 2019.** Thrombin and the coag-inflammatory nexus in neurotrauma, ALS, and other neurodegenerative disorders. *Frontiers in Neurology* **10**:59 DOI 10.3389/fneur.2019.00059.

**Ficiarà E, Munir Z, Boschi S, Caligiuri ME, Guiot C. 2021.** Alteration of iron concentration in Alzheimer's disease as a possible diagnostic biomarker unveiling ferroptosis. *International Journal of Molecular Sciences* **22(9)**:4479 DOI 10.3390/ijms22094479.

**François M, Karpe AV, Liu J-W, Beale DJ, Hor M, Hecker J, Faunt J, Maddison J, Johns S, Doecke JD. 2022.** Multi-omics, an integrated approach to identify novel blood biomarkers of Alzheimer's disease. *Metabolites* **12(10)**:949 DOI 10.3390/metabo12100949.

**Geroldi D, Minoretti P, Bianchi M, Di Vito C, Reino M, Bertona M, Emanuele E. 2005.** Genetic association of alpha2-Heremans-Schmid glycoprotein polymorphism with late-onset Alzheimer's disease in Italians. *Neuroscience Letters* **386(3)**:176–178 DOI 10.1016/j.neulet.2005.06.014.

**Goodman AB. 2006.** Retinoid receptors, transporters, and metabolizers as therapeutic targets in late onset Alzheimer disease. *Journal of Cellular Physiology* **209(3)**:598–603 DOI 10.1002/jcp.20784.

**Guan J, Wang P, Lu L, Zhao G. 2020.** Association of plasma transferrin with cognitive decline in patients with mild cognitive impairment and Alzheimer's disease. *Frontiers in Aging Neuroscience* **12**:38 DOI 10.3389/fnagi.2020.00038.

**Jeon SG, Cha M-Y, Kim J-I, Hwang TW, Kim KA, Kim TH, Song KC, Kim J-J, Moon M. 2019.** Vitamin D-binding protein-loaded PLGA nanoparticles suppress Alzheimer's disease-related pathology in 5XFAD mice. *Nanomedicine: Nanotechnology, Biology and Medicine* **17**:297–307 DOI 10.1016/j.nano.2019.02.004.

**Jiang R, Smailovic U, Haytural H, Tijms BM, Li H, Haret RM, Shevchenko G, Chen G, Abelein A, Gobom J. 2022.** Increased CSF-decorin predicts brain pathological changes driven by Alzheimer's Aβ amyloidosis. *Acta Neuropathologica Communications* **10**:1–17 DOI 10.1186/s40478-022-01398-5.

**Khoonsari PE, Shevchenko G, Herman S, Remnestål J, Giedraitis V, Brundin R, Degerman Gunnarsson M, Kilander L, Zetterberg H, Nilsson P. 2019.** Improved differential diagnosis of Alzheimer's disease by integrating ELISA and mass spectrometry-based cerebrospinal fluid biomarkers. *Journal of Alzheimer's Disease* **67(2)**:639–651 DOI 10.3233/JAD-180855.

**Kitamura Y, Usami R, Ichihara S, Kida H, Satoh M, Tomimoto H, Murata M, Oikawa S. 2017.** Plasma protein profiling for potential biomarkers in the early diagnosis of Alzheimer's disease. *Neurological Research* **39(3)**:231–238 DOI 10.1080/01616412.2017.1281195.

**Kononikhin AS, Zakharova NV, Semenov SD, Bugrova AE, Brzhozovskiy AG, Indeykina MI, Fedorova YB, Kolykhalov IV, Strelnikova PA, Ikonnikova AY. 2022.** Prognosis of Alzheimer's

disease using quantitative mass spectrometry of human blood plasma proteins and machine learning. *International Journal of Molecular Sciences* **23(14)**:7907 DOI 10.3390/ijms23147907.

**Lin Q, Cao Y, Gao J. 2015.** Decreased expression of the APOA1-APOC3–APOA4 gene cluster is associated with risk of Alzheimer's disease. *Drug Design, Development and Therapy* **9**:5421 DOI 10.2147/DDDT.S89279.

**Llano DA, Devanarayan V, Simon AJ, Alzheimer's Disease Neuroimaging Initiative. 2013.** Evaluation of plasma proteomic data for Alzheimer disease state classification and for the prediction of progression from mild cognitive impairment to Alzheimer disease. *Alzheimer Disease & Associated Disorders* **27(3)**:233–243 DOI 10.1097/WAD.0b013e31826d597a.

**Mohamed WA, Salama RM, Schaalan MF. 2019.** A pilot study on the effect of lactoferrin on Alzheimer's disease pathological sequelae: impact of the p-Akt/PTEN pathway. *Biomedicine & Pharmacotherapy* **111(Pt A)**:714–723 DOI 10.1016/j.biopha.2018.12.118.

**Moon M, Song H, Hong H, Nam D, Cha M, Oh M, Yu J, Ryu H, Mook-Jung I. 2013.** Vitamin D-binding protein interacts with Aβ and suppresses Aβ-mediated pathology. *Cell Death & Differentiation* **20(4)**:630–638 DOI 10.1038/cdd.2012.161.

**Nielsen JE, Honoré B, Vestergård K, Maltesen RG, Christiansen G, Bøge AU, Kristensen SR, Pedersen S. 2021.** Shotgun-based proteomics of extracellular vesicles in Alzheimer's disease reveals biomarkers involved in immunological and coagulation pathways. *Scientific Reports* **11(1)**:18518 DOI 10.1038/s41598-021-97969-y.

**Plucińska K, Dekeryte R, Koss D, Shearer K, Mody N, Whitfield PD, Doherty MK, Mingarelli M, Welch A, Riedel G. 2016.** Neuronal human BACE1 knockin induces systemic diabetes in mice. *Diabetologia* **59(7)**:1513–1523 DOI 10.1007/s00125-016-3960-1.

**Prokopenko D, Morgan SL, Mullin K, Hofmann O, Chapman B, Kirchner R, Alzheimer's Disease Neuroimaging Initiative, Amberkar S, Wohlers I, Lange C. 2021.** Whole-genome sequencing reveals new Alzheimer's disease–associated rare variants in loci related to synaptic function and neuronal development. *Alzheimer's & Dementia*

**Rehiman SH, Lim SM, Neoh CF, Majeed ABA, Chin A-V, Tan MP, Kamaruzzaman SB, Ramasamy K. 2020b.** Proteomics as a reliable approach for discovery of blood-based Alzheimer's disease biomarkers: a systematic review and meta-analysis. *Ageing Research Reviews* **60**:101066 DOI 10.1016/j.arr.2020.101066.

**Reseco L, Atienza M, Fernandez-Alvarez M, Carro E, Cantero JL. 2021.** Salivary lactoferrin is associated with cortical amyloid-beta load, cortical integrity, and memory in aging. *Alzheimer's Research & Therapy* **13(1)**:1–12 DOI 10.1186/s13195-021-00891-8.

**Rueli RH, Torres DJ, Dewing AS, Kiyohara AC, Barayuga SM, Bellinger MT, Uyehara-Lock JH, White LR, Moreira PI, Berry MJ. 2017.** Selenoprotein S reduces endoplasmic reticulum stress-induced phosphorylation of tau: potential role in selenate mitigation of tau pathology. *Journal of Alzheimer's Disease* **55(2)**:749–762 DOI 10.3233/JAD-151208.

**Shah A, Kishore U, Shastri A. 2021.** Complement system in Alzheimer's disease. *International Journal of Molecular Sciences* **22(24)**:13647 DOI 10.3390/ijms222413647.

**Shi X, Ohta Y, Liu X, Shang J, Morihara R, Nakano Y, Feng T, Huang Y, Sato K, Takemoto M. 2019a.** Acute anti-inflammatory markers ITIH4 and AHSG in mice brain of a novel Alzheimer's disease model. *Journal of Alzheimer's Disease* **68(4)**:1667–1675 DOI 10.3233/JAD-181218.

**Shi X, Ohta Y, Liu X, Shang J, Morihara R, Nakano Y, Feng T, Huang Y, Sato K, Takemoto M. 2019b.** Chronic cerebral hypoperfusion activates the coagulation and complement cascades in Alzheimer's Disease mice. *Neuroscience* **416(Suppl 1)**:126–136 DOI 10.1016/j.neuroscience.2019.07.050.

**Smith ER, Nilforooshan R, Weaving G, Tabet N. 2011.** Plasma fetuin-A is associated with the severity of cognitive impairment in mild-to-moderate Alzheimer's disease. *Journal of Alzheimer's Disease* **24(2)**:327–333 DOI 10.3233/JAD-2011-101872.

**Solovyev N. 2020.** Selenoprotein P and its potential role in Alzheimer's disease. *Hormones* **19(1)**:73–79 DOI 10.1007/s42000-019-00112-w.

**Song F, Poljak A, Kochan NA, Raftery M, Brodaty H, Smythe GA, Sachdev PS. 2014.** Plasma protein profiling of mild cognitive impairment and Alzheimer's disease using iTRAQ quantitative proteomics. *Proteome Science* **12(1)**:1–13 DOI 10.1186/1477-5956-12-5.

**Srinivasan K, Friedman BA, Etxeberria A, Huntley MA, van Der Brug MP, Foreman O, Paw JS, Modrusan Z, Beach TG, Serrano GE. 2020.** Alzheimer's patient microglia exhibit enhanced aging and unique transcriptional activation. *Cell Reports* **31(13)**:107843 DOI 10.1016/j.celrep.2020.107843.

**Suidan GL, Singh PK, Patel-Hett S, Chen Z-L, Volfson D, Yamamoto-Imoto H, Norris EH, Bell RD, Strickland S. 2018.** Abnormal clotting of the intrinsic/contact pathway in Alzheimer disease patients is related to cognitive ability. *Blood Advances* **2(9)**:954–963 DOI 10.1182/bloodadvances.2018017798.

**Tenner AJ. 2020.** Complement-mediated events in Alzheimer's disease: mechanisms and potential therapeutic targets. *The Journal of Immunology* **204**:306–315.

**Vowinkel T, Mori M, Krieglstein CF, Russell J, Saijo F, Bharwani S, Turnage RH, Davidson WS, Tso P, Granger DN. 2004.** Apolipoprotein A-IV inhibits experimental colitis. *The Journal of Clinical Investigation* **114**:260–269.

**Wright JW, Harding JW. 2015.** The brain hepatocyte growth factor/c-Met receptor system: a new target for the treatment of Alzheimer's disease. *Journal of Alzheimer's Disease* **45(4)**:985–1000 DOI 10.3233/JAD-142814.

**Yang M-H, Yang Y-H, Lu C-Y, Jong S-B, Chen L-J, Lin Y-F, Wu S-J, Chu P-Y, Chung T-W, Tyan Y-C. 2012.** Activity-dependent neuroprotector homeobox protein: a candidate protein identified in serum as diagnostic biomarker for Alzheimer's disease. *Journal of Proteomics* **75(12)**:3617–3629 DOI 10.1016/j.jprot.2012.04.017.

**Zhang Z-H, Song G-L. 2021.** Roles of selenoproteins in brain function and the potential mechanism of selenium in Alzheimer's disease. *Frontiers in Neuroscience* **15**:215 DOI 10.3389/fnins.2021.646518.

**Zhang P, Yang Z, Zhang C, Lu Z, Shi X, Zheng W, Wan C, Zhang D, Zheng C, Li S. 2003.** Association study between late-onset Alzheimer's disease and the transferrin gene polymorphisms in Chinese. *Neuroscience Letters* **349(3)**:209–211 DOI 10.1016/S0304-3940(03)00837-1.

**Zhu Y, Hilal S, Chai YL, Ikram MK, Venketasubramanian N, Chen CP, Lai MK. 2018.** Serum hepatocyte growth factor is associated with small vessel disease in Alzheimer's dementia. *Frontiers in Aging Neuroscience* **10**:8 DOI 10.3389/fnagi.2018.00008.