# Peer review of "A multi-ethnic proteomic profiling analysis in Alzheimer’s disease identifies the disparities in dysregulation of proteins and pathogenesis"

_PeerJ, doi:10.7717/peerj.17643_

## Round 0.1 · original submission · Major Revisions

Please address the concerns of both reviewers and amend the manuscript accordingly.

**Language Note:** The review process has identified that the English language must be improved. PeerJ can provide language editing services - please contact us at [email protected] for pricing (be sure to provide your manuscript number and title). Alternatively, you should make your own arrangements to improve the language quality and provide details in your response letter. – PeerJ Staff

Reviewer 1 ·

Basic reporting

The manuscript is clearly written using professional, unambiguous English throughout. The language used facilitates comprehension of this technical subject matter.

Experimental design

Study Design

(1)The use of a multi-ethnic cohort is a strength of the study, allowing investigation of biological and environmental influences on AD pathogenesis. However, the sample size is quite small. The authors should discuss the potential limitations of the small sample size and any efforts to minimize bias through their analysis approach.
(2)More details should be provided on the sample processing methodology to ensure adequate reproducibility. For example, provide more specifics on the depletion and digestion methods.
(3)The statistical methods used for the proteomics analysis are appropriate, but more details should be included on the differential expression analysis approach. Were any adjustments made given the small sample size?

Validity of the findings

The conclusions accurately summarize the key findings related to differences in dysregulated proteins and pathways between the Chinese and Malay cohorts. The authors refrain from overstating the conclusions based on this preliminary data.

Additional comments

(1)The discussion provides good interpretation of the pathways and protein interactions identified in the Chinese and Malay cohorts. The authors link the findings to previous literature well.
(2)More comparison of the differences and similarities found between the ethnic groups would further enhance the discussion. What might account for the variation in dysregulated proteins?
(3)Additional limitations should be noted, including the preliminary nature of these findings given the small sample size. Comment on need for validation in larger cohorts.
(4)Expand on potential next steps and future research directions to build on this study. Multi-omics approaches could provide further insight into mechanisms.
(5)The survey data showed significant differences between the Chinese and Malay groups in living arrangements and sweet food/drink consumption. The authors link the higher sweet consumption in Malays to dysregulation of APOA4, but more interpretation is needed. Is the living arrangement difference relevant?
(6)The meta-analysis provides support for some of the key protein findings (TF, AHSG, A1BG, APOA4, C4A). The authors could compare their effect sizes for these proteins to the aggregated effects. Does your data show stronger or weaker dysregulation?
(7)Many proteins identified have no known or unclear links to AD (e.g. ITIH1, HGFAC, A1BG, KRT1, KRT10). Further exploration of potential connections to pathways would be interesting. Are they linked to ethnicity-specific environmental factors?
(8)The study identified some pathway similarities but differences in specific proteins between ethnic groups. More could be discussed about potential genetic vs environmental contributors to the disparities.

Reviewer 2 ·

Basic reporting

no comment

Experimental design

no comment

Validity of the findings

no comment

Additional comments

In this manuscript, authors have conducted a proteomic profiling analysis to find differentially expressed proteins between matched control and Alzheimer’s affected study subjects in two distinct ethnic backgrounds: Chinese and Malays. Accordingly, DEPs were found in both groups but with no overlap. Simultaneously, authors have analyzed the protein-protein interaction networks of DEPs and their relevant functional pathways.

Major limitation of the current study is small sample size which undermines the findings presented in the report.
Gender based analysis could have been done with a higher sample size.
Did authors analyzed DEPs between control and AD patients by combining the data from both ethnic groups?
Lines 190: Authors ranked top proteins based on adjusted p-value of <0.05. What is adjusted p-value? Is it based on multiple testing?
Lines 269-270: “Figure 3a shows the.... CADvC and MADvC” are repetition of lines 262-263: “A Venn diagram.....is shown in Figure 3A”. Similarly, lines 278-279: “The PPI networks....in Figure 4” with that of lines 281-282: “Figure 4 illustrates...and MADvC”.
The result section should be divided into subsections such as 1. Differential abundance of proteins, 2. Protein-protein interaction network analysis 3. Functional analysis and 4. Meta-analysis
Authors should seek professional editorial help for correcting grammatical errors present throughout the manuscript.

---

## Round 0.2 · Major Revisions

Please address the issues pointed out by the reviewer and amend your manuscript accordingly.

**Language Note:** The review process has identified that the English language must be improved. PeerJ can provide language editing services - please contact us at [email protected] for pricing (be sure to provide your manuscript number and title). Alternatively, you should make your own arrangements to improve the language quality and provide details in your response letter. – PeerJ Staff

Reviewer 1 ·

Basic reporting

The manuscript is generally well-written in professional English. However, there are some grammatical errors and expressions that need to be corrected to further improve the clarity. I suggest the authors carefully proofread the manuscript and polish the language.

Experimental design

The research question of investigating ethnic differences in AD blood proteome between Chinese and Malay populations is well-defined and relevant, as it may provide new insights into the pathogenesis of AD and the potential impact of environmental and genetic factors. However, I suggest the authors further elaborate on how this study fills the knowledge gap in the field. The introduction should clearly state the limitations of previous studies and highlight the novelty and significance of the present work.

Validity of the findings

The study provides new insights into the ethnic differences in AD blood proteome between Chinese and Malay populations in Malaysia. The identification of differentially expressed proteins and the associated biological pathways in each ethnic group is novel and may contribute to a better understanding of the pathogenesis of AD. However, given the small sample size and the lack of replication cohort, the impact and generalizability of the findings may be limited. I suggest the authors clearly state the rationale and potential benefits of replicating the findings in larger and independent cohorts, which will strengthen the validity and clinical significance of the study.

Additional comments

I suggest the authors further clarify the importance of studying protein expression differences between different ethnicities for understanding the pathogenesis of AD in the introduction section.

The current research gap needs to be supplemented, and the necessity and significance of studying ethnic differences between Malaysian Chinese and Malays should be explained.

I suggest the authors provide detailed inclusion and exclusion criteria for AD patients and normal controls included in the study in the methods section, especially how to exclude the influence of other types of dementia and comorbidities.

I suggest the authors provide more details on the methods of proteomic analysis, including the model of mass spectrometer used, database search, and quantitative analysis parameter settings, to ensure the repeatability of the experiment.

I suggest the authors describe the statistical analysis methods for the questionnaire survey data in more detail, including the specific statistical tests used and the setting of statistical significance levels.

I suggest the authors describe the study screening and data extraction process for the meta-analysis in more detail, including the databases searched, search strategy, inclusion and exclusion criteria, etc. At the same time, the heterogeneity of the included studies needs to be assessed.

I suggest the authors discuss the results of PCA and hierarchical clustering more fully in the results section, and explain the biological significance of these results.

I suggest the authors further explore the role and mechanism of key differentially expressed proteins between the two ethnicities, such as VDBP and APOA4, in the pathogenesis of AD and their association with environmental risk factors in the discussion section.

I suggest the authors conduct an integrated analysis of the questionnaire survey of environmental risk factors with the proteomic results, to clarify the possible mechanisms by which environmental factors lead to AD by influencing protein expression, although a questionnaire survey was conducted.

I suggest the authors compare the results of this study with previous proteomic studies more comprehensively in the discussion section, especially the differences between different ethnicities, and explore the potential mechanisms. The current discussion is not deep enough.

I suggest the authors expand the sample size for validation studies in the future, given the small sample size. Some in vitro experiments can be supplemented, such as absolute quantification of proteins, cell or animal validation experiments of key pathways predicted by bioinformatics analysis, etc., to support the results of this study.

---

## Round 0.3 · accepted · Accept

Comments of the reviewers were addressed and revised manuscript is acceptable now

Reviewer 2 ·

Basic reporting

Authors have addressed the reviewer comments satisfactorily.

Experimental design

NA

Validity of the findings

NA

Additional comments

NA